# Migration of the Lag Screw after Intramedullary Treatment of AO/OTA 31.A2.1-3 Pertrochanteric Fractures Does Not Result in Higher Incidence of Cut-Outs, Regardless of Which Implant Was Used: A Comparison of Gamma Nail with and without U-Blade (RC) Lag Screw and Proximal Femur Nail Antirotation (PFNA)

**DOI:** 10.3390/jcm8050615

**Published:** 2019-05-07

**Authors:** Nikolaus Wilhelm Lang, Robert Breuer, Hannes Beiglboeck, Alexandru Munteanu, Stefan Hajdu, Reinhard Windhager, Harald Kurt Widhalm

**Affiliations:** 1Department of Orthopedics and Traumatology, Clinical Division of Traumatology, Medical University of Vienna, 1090 Vienna, Austria; nikolaus.lang@meduniwien.ac.at (N.W.L.); robert.breuer@meduniwien.ac.at (R.B.); hannes.beiglboeck@hotmail.com (H.B.); stefan.hajdu@meduniwien.ac.at (S.H.); 2Department: Medical School, University College London, London WC1E 6BT, UK; alexandru.munteanu.14@ucl.ac.uk; 3Department of Orthopedics and Traumatology, Clinical Division of Orthopedics, Medical University of Vienna, 1090 Vienna, Austria; reinhard.windhager@meduniwien.ac.at

**Keywords:** AO/OTA 31.A2.1-3 femur fractures, cut-out, Proximal Femur Nail Antirotation (PFNA), Gamma Nail, U-Blade RC lag-screw

## Abstract

The best intramedullary nail for the treatment of unstable AO/OTA 31.A2.1-3 fractures remains uncertain. A total of 237 patients (45 male, 192 female) were eligible for the assessment with an average age of 81.9 ± 10.5 years and a minimum follow-up of six months. We assessed the cut-out rate, the TAD and calTAD (Tip Apex distance) of three different implants. An overall cut-out rate of 2.5% (*n* = 6) was observed. The cut-out rate was 1.2% (*n* = 1) in the Proximal Femur Nail Antirotation (PFNA) group, 3.7% (*n* = 3) in the Gamma Nail group and 2.9% (*n* = 2) in the Gamma3^®^ with a U-Blade (RC) lag-screw group. The TAD and calTAD differed between the cut-out and non-cut group—20.0 mm vs. 18.5 mm and 13.1 mm vs. 15.3 mm, respectively. A significantly higher TAD of 32.5 mm could be seen in the cut-out after PFNA (*p* < 0.0001). The only significant change in follow-up using Parker’s ratio was observed in the PFNA group (*p* < 0.0001). The rate of patients requiring surgery after complications was 8.4% (*n* = 20) without any significant difference between the three groups. The PFNA blade showed significant migration within the femoral head, however the cut-out rate remained the smallest compared to Gamma3 with or without additional U-Blade (RC) lag screw.

## 1. Introduction

Trochanteric femoral fractures are one of the most common injuries in the elderly population and the incidence will only increase due to demographic changes [1]. An optimal operative treatment is immanent regarding the typical geriatric patient, benefiting from early mobilization and weight bearing in order to avoid immobility-related complications [1,2]. Additionally, the enormous socioeconomic impact has to be considered [1,2,3,4]. The best fixation method remains debated, with an overall tendency to use cephalomedullary implants [2,4,5] especially in unstable fracture patterns where the implant’s mechanical properties are superior to extramedullary systems [6,7,8,9]. The Gamma Nail ^®^ (Stryker Trauma, Murnau, Germany) and Proximal Femur Nail Antirotation (PFNA)^®^ (DePuy Synthes, Umkirch, Germany) are the most commonly used intramedullary nails for the treatment of pertrochanteric fractures. For both implants, the migration of the femoral head into varus and retroversion, and the subsequent cut-out of the lag screw/blade, is the most common mechanical complication and are described between 0–7% [10,11,12]. The compromised bone quality in the head and neck of the femur in the typical geriatric patient with osteoporotic bone changes requires an exact positioning of the head screw in the center-center or low-center position and, preferably, a small tip-apex distance (TAD) [13,14]. Additionally, in relation to patient-related characteristics and the positioning of the lag screws, the design of the implants must be taken into account. Non-cylindrical and more blade-like head fixation devices have been developed to prevent, or at least reduce, the failure rate in the latest implant generation [15,16]. None of these implants have yet shown superiority regarding prevention of undesired rotational movement and consequent cut-out [17,18,19]. To the best of our knowledge, current literature lacks direct comparisons of clinical and radiological results between the latest most used implant technologies, especially in unstable AO/OTA 31. A2.1-3 femur fractures.

The aim of the study was to assess (1) the cut-out rate, (2) migration of the lag screw/ blade, and (3) implant failure in geriatric patients with unstable AO/OTA 31.A2.1-3 fractures treated with PFNA^®^, Gamma3^®^ or Gamma3^®^ with U-Blade (RC) lag screw.

## 2. Materials and Methods

This study was approved by the local ethic committee and carried out according to the declaration of Helsinki (Ethical Commission Number 1485/2013).

Inclusion criteria for this study were: unstable OTA/AO 31A2.1-3 femur fractures treated with Gamma3^®^ Nail (Stryker Trauma, Umkirch, Germany), either with standard lag screws or with the U-Blade (RC) Lag Screw (Stryker Trauma, Umkirch, Germany), or Proximal Femur Nail Antirotation (PFNA^®^, Depuy Synthes, Umkirch, Germany).

Patients with pathological fractures as well as patients treated with a long cephalomedullary nail were excluded. Data was retrieved from our department’s database and completed by chart reviews. All fractures were classified by two of the authors using the AO/OTA system [20] (Nikolaus W. Lang and Harald K. Widhalm). Minimum follow up was six months, when general fracture healing was obtained; the mean follow up was 10.8 ± 4.1 months.

A total of 237 patients were eligible for the assessment with a mean age of 81.9 ± 10.5 years and a minimum follow-up of six months. A total of 82 patients were treated with a standard Gamma3^®^(Gamma), 69 patients with a Gamma3^®^ with an additional U-Blade (RC) lag-screw (U-Blade) and 86 patients were treated with a PFNA^®^ (PFNA).

### 2.1. Diagnosis, Surgical Technique and Aftercare

All fractures were diagnosed by standard radiographs. Where possible, the surgery took place within 24–48 h of the initial trauma, in line with current literature, to decrease the perioperative risks and mortality [21,22,23,24].

In all groups, the mean time between initial trauma and surgery, using general or spinal anesthesia, was 1.4 ± 1.3 days. Mean BMI (body mass index) of the patients was 24.2 ± 3.2 and the median American Society of Anesthesiologists (ASA) Score was 2. Patients were positioned supine on the traction table with their uninjured leg in a leg holder. Patients either received a standard 200 × 11 mm Gamma3^®^ nail with standard lag screw with/with-out additional antirotation U-Blade (RC) lag-screw or they received a standard 200 × 11 mm PFNA^®^ with a blade in appropriate length.

Implantation of a cephalomedullary nail was performed under dual-plane fluoroscopy using two C-arms, simultaneously. One C-arm was placed for an Anterior Posterior (AP)-view while the second provided a lateral view in neutral position. Intraoperative alterations of the angle of the lateral C-arm, up to 25°, were performed depending on the fracture and morphology of the femur.

According to our guidelines, the position of the lag screw should be in the caudal third of the antero-posterior plane and in the middle of the lateral plane of the femoral head (Figure 1).

Mobilization with full weight-bearing using two crutches or wheeled walker started on day 1 post-operatively under the instruction of experienced physiotherapists. The radiographs in the two planes were repeated on day 5.

### 2.2. Outcome Parameters

The caput–collum–diaphyseal (CCD) angle was measured on the contralateral side. Cut-out was defined as extrusion of the lag screw by more than 1 mm from the femoral head [25]. The modified Parker’s ratio was used to categorize the placement of the lag screw in the femoral head in both the antero-posterior and lateral plane [25,26,27]. To identify lag screw position and migration, as well as rotation of the femoral head, two observers (LNW, WHK) assessed the Parker’s ratio on intraoperative radiographs and at 6 and 12 months post-operatively [25] (Figure 2). The likelihood of cut-out was assessed by the calcar referenced tip–apex distance (calTAD) and the tip–apex distance (TAD) on postoperative radiographs [28] (Figure 2 and Figure 3). In addition, we evaluated the lateral overhang of the lag screw/blade from the lateral cortices of the trochanteric bone as a percent of the total length of the lag screw/blade. The Parker mobility score was used to assess all patients’ mobility pre-operatively, at discharge from hospital, and at the last follow-up to quantify and compare clinical outcomes between groups [29]. Furthermore, we analyzed the mean operation time as well as the mean hospital stay.

### 2.3. Complications

All implant-related failures, such as nail breakage and lateralization, migration, penetration or cut-out of the lag screw were reported. Additionally, we evaluated all patient-related complications including intraoperative adverse events (cardiovascular failure), infection, and thrombosis.

### 2.4. Statistical Analysis

Quantitative data was compared between the three groups or within the single groups (standard lag screw/U-Blade (RC) lag screw, PFNA blade) using a one-way ANOVA, the Student’s *t*-test or the Mann–Whitney U-test. Qualitative data analyses were performed utilizing the χ²-analysis. Statistical significance was set at α = 0.05. Multiple regression analysis with a 95% confidence interval was used to examine the independent associations of various demographic and injury-related factors.

## 3. Results

All implants showed low cut-out rates with 1.2% (*n* = 1) in the PFNA group and 2.9% (*n* = 2) and 3.7% (*n* = 3) in the Gamma group and U-blade group, respectively. These differences, however, lacked statistical significance. Both the mean TAD and calTAD among those who suffered from cut-out was higher than those who were spared this complication—20.0 mm and 15.3 mm vs 18.5 mm and 13.1 mm, respectively. Neither of these differences achieved statistical significance (*p* > 0.05)

However, in a subgroup analysis of patients suffering a cut-out, the patient in the PFNA group was the only one that showed a significantly higher TAD (32.5 mm *p* < 0.0001), while the patients of the Gamma nail group, with or without U-blade, showed no significant difference. (Table 1 and Table 2)

Demographic parameters, including age, gender, BMI, ASA score, lateral overhang of the lag screw/blade and time of surgery, showed no statistically significant correlation to cut-out rates and risk of complications (*p* > 0.05).

### 3.1. Time to Cut Out/Treatment

The single cut-out after osteosynthesis with a PFNA happened six weeks post-operatively. The revision surgery involved the removal of the PFNA and implantation of a Gamma nail. A central cut-out occurred three weeks later and the patient had to undergo a total hip arthroplasty. Despite adequate fracture reduction, poor bone quality and suboptimal placement of the blade in the femoral head-neck fragment led to the cut-out (Figure 4). Postoperative mobilization was performed according to the standard protocol of our department, starting with full weight-bearing at the second day post-surgery.

In the Gamma nail population, one cut-out was observed three weeks post-primary implantation. The patient had a revision endoprosthesis (Helios). The second cut-out happened 16 weeks postoperatively with the subsequent revision surgery replacing the lag screw with a shorter version. A non-union developed over the next seven months requiring implant removal followed by a hemiarthroplasty. The third patient, suffering a cut-out seven weeks post-primary intervention, was revised with a hemiarthroplasty.

Among those treated with Gamma nail + U-Blade, the first cut-out happened 14 weeks post-implantation. The initial revision used a long PFNA nail, but following a second cut-out diagnosed two years later, the patient had to undergo a hemiarthroplasty. A second patient suffered from cut-out 14 months post-operatively, complicated by concomitant femoral head necrosis, requiring revision with a hemiarthroplasty (Table 3).

### 3.2. Migration of the Lag Screw/Blade

We identified lateralization of the blade in three patients (3.5%) in the PFNA group, give patients (6.1%) in the Gamma nail group, and 10 patients (14.5%) in the Gamma nail with U-blade group. These differences, however, were not statistically significant, nor was there any relationship between lateralization of the lag screw/blade and cut-out rates.

Nevertheless, follow-up demonstrated a significant change in Parker’s ratio within the PFNA group (*p* < 0.0001). Follow-up of the Gamma and U-Blade group detected only a marginal migration of the lag screw. Comparing the PFNA group with the Gamma and the U-Blade group, we found a significant difference in migration of the blade/screw within the femoral head and neck fragment. (*p* < 0.0001). No correlation was observed between CalTAD, TAD, and Parker’s ratio among the patients with cut-outs in the Gamma and the U-Blade group. Using Parker’s ratio as predictor of blade/screw migration only shows significance in the single cut-out post-PFNA implantation.

Overall, we observed a tendency for higher complication rates among the gamma group, but this lacked statistical significance (*p* = n.s.) (Table 4).

Mean overhang on the lateral cortical bone was 10% ± 3% of the total lag screw/blade. There was no relationship between the length of the overhang and lateralization or migration of the lag screw/blade.

### 3.3. Functional Outcome

As anticipated, the mean Parker mobility score for all three groups, compared to preoperatively, decreased at discharge from hospital (7.1 ± 2.9 vs. 5.2 ± 2.8, respectively; *p* < 0.0001), but rebounded at the last follow-up appointment (6.3 ± 2.7). The mean time in hospital was 15.7 ± 7.2 days. There was no significant difference between the three groups either in mobility outcomes or length of stay (*p* > 0.05).

## 4. Discussion

The most important finding of the study is that the cross-group cut-out rate is very low (3%). The PFNA showed significant (*p* < 0.0001) post-operative migration of the blade in the femoral head/neck as shown by radiographs taken on day 5 and six months post-operatively. However, this had no impact on the risk of cut-out. In contrast, the standard Gamma nail group showed a tendency (*p* = n.s.) for higher number of complications leading to revision surgery.

In the direct comparison between three frequently used implants, we investigated the differences between the latest generation of cephalomedullary implants in the care of pertrochanteric fractures, especially relating to complications and adverse effects. The principle of intramedullary force conduction is a well-established treatment option, especially in unstable fracture patterns [2,5,9]. Despite the technical evolution regarding implant designs, the problem of screw/blade migration, or even cut-out, still remains [10,11,12].

Cut-out rates for the PFNA and Gamma nail are reported between 0%–6.2% and 1.85%–6.7% [12,30], respectively. However, data on unstable AO/OTA31.A2 1-3 fractures is still inconclusive.

Both the specific, as well as the overall, cut-out rates in this series concur with current literature. Various methods have been developed to assess the optimal placement of the lag screw and therefore minimize the risk of cut-out [27,28,31,32]. TAD and Parker’s ratio showed to be the most accurate predictors of this complication [27,32]. According to Kashigar et al., who introduced the calTAD, this parameter should surpass the TAD in predictive value [28]. In our survey, we only measured one statistically increased TAD in the single case of cut-out in the PFNA group. All other cut-outs, as well as the overall cut-out number, were not related to higher values in TAD, calTAD, or Parker’s ratio.

Many different manufacturers provide implants with various characteristics. When driving in the blade of the PFNA, the cancellous bone should be compressed and compacted to increase implant stability. This is in contrast to conventional lag screw systems, especially in osteoporotic bone where there is a loss of bone density and stability. Furthermore, the diameter of the PFNA blade is 20% larger than the lag screw of the Gamma nail, intending to decrease failure and cut-out rates particularly in osteoporotic bone. Interestingly, only the PFNA group showed a significant change in Parker’s ratio between the initial value and the time of follow up. This hints at an increased migration rate of the blade without a higher cut-out rate. By the means of its lag screw, the Gamma nail offers the possibility of fracture compression. We suspect that this results in higher primary stability compared to PFNA, which can be deduced from a constant Parker’s ratio over time. Nevertheless, we observed a higher cut-out rate in the Gamma nail group.

The Gamma nail allows for the replacement of the standard lag screw with a U-blade lag screw set—a combination of the lag screw with a U-shaped clip increasing the diameter by 2 mm. The resulting increase in surface area, by about 15%, improves stability against rotation and cut-out [18]; indicated in unstable fracture patterns and highly osteoporotic bone. Concordant to the only paper examining the effects of this modification relating to a standard Gamma nail [18], other than a tendency to a reduced cut-out rate (2.9% vs. 3.7%, *p* = n.s.), we found no significant difference in outcomes using this gadget. However, the technique required to implant a Gamma nail with U-blade significantly prolonged the operating time (57.1 ± 25.3 min vs. 67.8 ± 23.8 min; *p* < 0.0040). Furthermore, the number of screw lateralization was increased in comparison to the other implants (3.5% vs. 6.1% vs. 14.5% *p* = n.s.) but this lacked statistical significance.

In the PFNA group, in keeping with current data, no cases of non-union were evident [9,19,33]. Lateralization was observed in 3.5%, conforming to previously described rates ranging from 0.6% [33] to 2.6% [34] and 5.1% [35]. The possibility of cement augmentation, especially in high risk patients, has already been described as a possible solution to avoid potential blade migration [36].

For the latest generation of the Gamma nails, complication rates around 7% are reported in the literature; these include non-unions, nail breakage, breakage of the distal screw, secondary femoral fracture, and loss of reduction [30]. In our series, we observed a similar complication rate (4.6%). Of note, three cases of nail breakage occurred which is described as a very rare event in literature (1.3% vs. 0.2–5.7%, respectively) [37].

As of yet, there is no data regarding implant failure for the Gamma nail in combination with the U-blade. We observed one non-union as well as one secondary femoral fracture. This hints towards a lower rate of implant failure than the standard Gamma nail, but not statistically significant. Contrastingly, a higher rate of lag screw lateralization occurred, due to varus collapse of the fracture, without further migration within the femoral head/neck. This might result from incorrectly performing the surgery, such as no tightening the set-screw at the end of the surgery.

Several factors are associated with failure of cephalomedullary nailing for pertrochanteric fractures [38,39]. Recent literature identified no relation between patient demographics and implant failure. However, increased TAD and unstable, or poorly reduced, fractures were found to be the principal causes of implant cut-out and failure (REF) [38,39]. In keeping with this data, the single PFNA cut-out had a significantly higher TAD. Contrastingly, adequate fracture reduction was achieved in all our patients who subsequently suffered a cut-out.

### Limitations

There are several limitations to this study. Firstly, its retrospective design did not eliminate a selection bias and prevented additional dual energy x-ray absorptiometry (DEXA) scans from being performed. Secondly, the choice of the implant was at the decision of the attending surgeon. Thirdly, the small sample size might underrepresent the true cut-out rate. Finally, the assessment, and consequently the optimal treatment, of rotational instability in OTA/AO 31.A2.1-3 femur fractures still remains difficult and uncertain in some cases.

## 5. Conclusions

The PFNA blade showed significant migration within the femoral head, however the cut-out rate was the smallest (1.2%) compared to Gamma3 and Gamma3 with additional U-Blade RC lag-screw. Due to their 20% larger diameter, the PFNA blade and Gamma with U-Blade RC lag-screw seem to be appropriate solutions for patients with osteoporotic bone and unstable AO/OTA 31.A2.1-3 fractures.

## Figures and Tables

**Figure 1 jcm-08-00615-f001:**
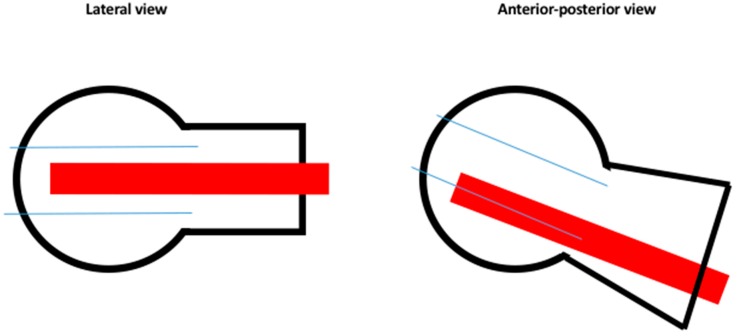
Aimed lag screw/blade position. The aimed position of the lag screw/blade in the femoral head and neck fragment according to departmental guidelines.

**Figure 2 jcm-08-00615-f002:**
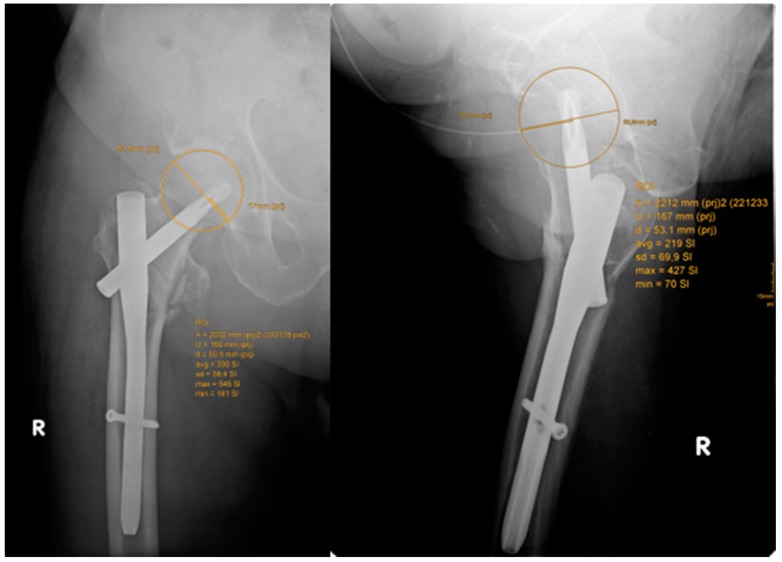
Assessment of the Parker’s ratio anterior–posterior view and lateral view.

**Figure 3 jcm-08-00615-f003:**
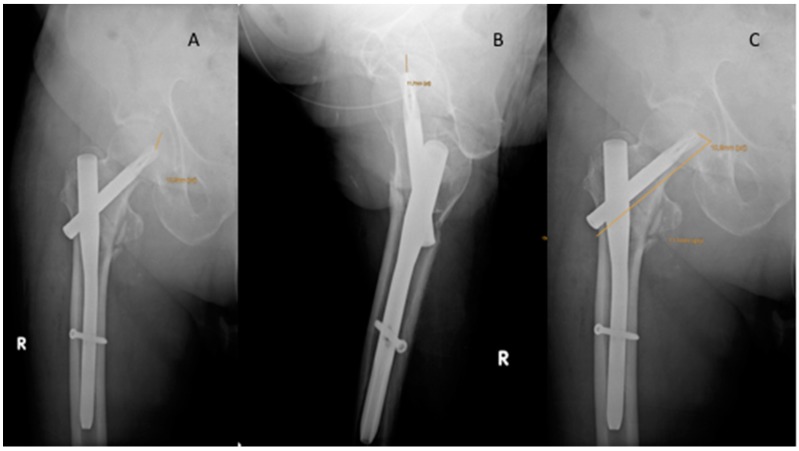
Assessment of the tip–apex distance (TAD) and of the calculated tip–apex distance (calTAD). Assessment of the TAD anterior–posterior view and lateral view (**A**,**B**) according to Baumgartner et al. [27], assessment of the calculated calTAD anterior–posterior view (**C**).

**Figure 4 jcm-08-00615-f004:**
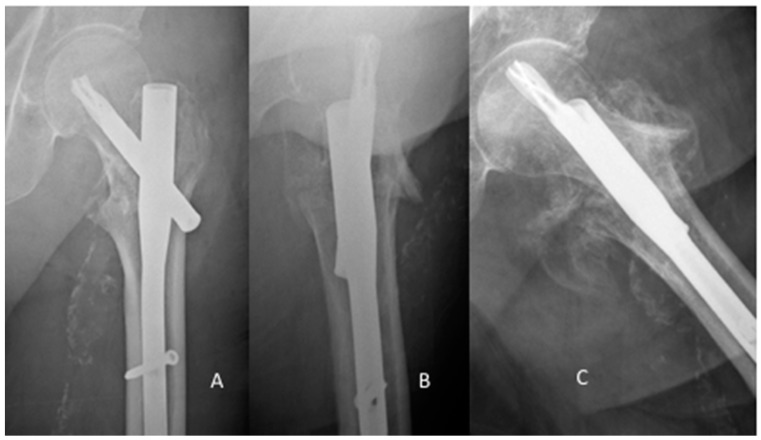
Cut-out Proximal Femur Nail Antirotation (PFNA). PFNA AP view and lateral view post surgery (**A**,**B**), imminent cut-out in lateral view six weeks post-surgery (**C**).

**Table 1 jcm-08-00615-t001:** Patients’ demographics and overall general complications.

	PFNA	Gamma	U-Blade	Total
	(*n* = 86)	(*n* = 82)	(*n* = 69)	(*n* = 237)
Male/Female	15/71	19/63	11/58	45/192
Age (years)	80.7±11.7	82.0 ± 10.0	83.2 ± 9.6	81.9 ± 10.5
BMI (kg/m^2^)	23.4 ± 3.8	23.9 ± 4.6	24.6 ± 4.0	24.0 ± 4.1
Operation (min)	56.5 ± 22.4	57.1 ± 25.3	67.8 ± 23.8 *	60.5 ± 23.8
Lateralization	3	5	10	22
Cut-out	1	3	2	6
Infection (superficial)	2	2	0	4
Complications requiring surgery	4 (4.7%)	12 (14.6%)	4 (5.7%)	8.4%

* *p* < 0.0040, significant longer operation time for U-Blade group compared to PFNA and Gamma group.

**Table 2 jcm-08-00615-t002:** Lag screw/blade position in the femoral head and neck fragment post-implantation and at latest follow-up.

	PFNA	Gamma	U-Blade	Total
Parker’s ratio AP	45.1 ± 8.0	43.2 ± 8.1	47.8 ± 6.6	45.3 ± 7.8
Parker’s ratio AP f/u	54.6 ± 8.7 *	45.1 ± 8.3	49.2 ± 7.7	49.8 ± 8.9
Parker’s ratio LAT	54.3 ± 7.7	55.2 ± 8.0	50.8 ± 7.2	53.6 ± 7.8
Parker’s ratio LAT f/u	53.7 ± 8.4	55.9 ± 7.1	49.3 ± 6.4	53.0 ± 7.9
CalTAD	12.6 ± 2.9	12.5 ± 2.9	14.2 ± 3.6	13.3 ± 3.3
TAD	20.4 ± 5.0	18.2 ± 4.8	16.5 ± 3.9	18.5 ± 4.8
Lateral overhang	14%	8%	8%	10%

* *p* < 0.0001 significant difference between postoperative Parker’s ratio AP and Parker’s ratio f/u concluding significant migration of the blade.

**Table 3 jcm-08-00615-t003:** Lag screw/blade position in the femoral head and neck fragment of patients sustaining a cut-out.

	PFNA	Gamma	U-Blade	Total
Number	1	3	2	6
Parker’s ratio AP	39	48.7 ± 5.6	50.5 ± 3.5	47.9 ± 5.8
Parker’s ratio LAT	67	58.3 ± 7.6	60.5 ± 4.9	56.6 ± 11.6
CalTAD	18.7	14.6 ± 1.4	15.0 ± 1.4	15.3 ± 1.9
TAD	32.8 *	18.3 ± 5.9	17.1 ± 4.3	20.0 ± 7.3
Lateral overhang	6%	11%	7%	9%

* *p* < 0.0001, significantly higher TAD compared to the non cut-out patients in the PFNA group.

**Table 4 jcm-08-00615-t004:** Complications requiring surgical revision.

	PFNA	Gamma	U-Blade	Total
Implant breakage		3		3
lateralization	1	2		3
Wound infection	1 *	1 *		2
Secondary femur fracture		1	1	2
Non-union			1	1
Overall %	2.3%	8.5%	2.9%	4.6%

* Local debridement no implant removal necessary.

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
