# Peer review of "Migration of the Lag Screw after Intramedullary Treatment of AO/OTA 31.A2.1-3 Pertrochanteric Fractures Does Not Result in Higher Incidence of Cut-Outs, Regardless of Which Implant Was Used: A Comparison of Gamma Nail with and without U-Blade (RC) Lag Screw and Proximal Femur Nail Antirotation (PFNA)"

_jcm, 2019, doi:10.3390/jcm8050615_

Reviewer 1 Report

Comments on Manuscript Number: jcm-486287

Title: Migration of the lag screw after intramedullary treatment of AO 31.A2 pertrochanteric fractures does not result in higher incidence of cut-outs, regardless of which implant was used. A comparison of Gamma Nail with and without U-Blade and PFNA.

 The authors present their study to assess cut-out rates, amount of migration of the lag screw/blade, and implant failure in geriatric patients undergoing fixation of trochanteric fractures by different IM nails.

The paper is highly interesting and substantially well written and organized, but some points have to be addressed before considering it viable for publication.

·    Comments:

- English style is good but several terms have to be changed: for instance, “non union” is better than “pseudoarthrosis”

- One of the debated point in the scientific community of orthopaedic surgeons is the most appropriate lateral (oblique) view of the head/neck of femur during surgery. Please, provide specifically the authors’ point of view regarding the alignment of the C-arm with respect to the patient during the procedure

- It would be crucial for the readers to report the preop and postop x-rays of the failed cases, mostly to understand why PFNa nail failed in the single patient and the actual migration rate: this is the most important finding of the study and images would be dramatically clear. Would a suboptimal reduction the cause of PFNa failure? Was a too early mobilization or weight bearing concession the actual cause of failure of the fixation? Without a direct readers’ evaluation the conclusion is only up to the authors

- One of the very first and to date few clinical papers on the use of modern generation nails in such fractures should be referred to and reported on references:

Carulli C, Piacentini F, Paoli T, Civinini R, Innocenti M. A comparison of two fixation methods for femoral trochanteric fractures: a new generation intramedullary system vs sliding hip screw. Clin Cases Miner Bone Metab. 2017 Jan-Apr;14(1):40-47. doi: 10.11138/ccmbm/2017.14.1.040. Epub 2017 May 30. PMID: 28740524

Author Response

- English style is good but several terms have to be changed: for instance, “non union” is better than “pseudoarthrosis”

àchanges were made accordingly

- One of the debated point in the scientific community of orthopaedic surgeons is the most appropriate lateral (oblique) view of the head/neck of femur during surgery. Please, provide specifically the authors’ point of view regarding the alignment of the C-arm with respect to the patient during the procedure

àImplantation of a cephalomedullarynail was performed under dual-plane fluoroscopy using two C-arms, simultaneously. One C-arm was placed for an AP-view while the second provided a lateral view in neutral position. Intraoperative alterations of the angle of the lateral C-arm, up to 25°, were performed depending on the fracture and morphology of the femur.

- It would be crucial for the readers to report the preop and postop x-rays of the failed cases, mostly to understand why PFNa nail failed in the single patient and the actual migration rate: this is the most important finding of the study and images would be dramatically clear. Would a suboptimal reduction the cause of PFNa failure? Was a too early mobilization or weight bearing concession the actual cause of failure of the fixation? Without a direct readers’ evaluation the conclusion is only up to the authors

àThe single cut-out after osteosynthesis with a PFNA happened 6 weeks post-operatively. Despite adequate fracture reduction, poor bone quality and suboptimal placement of the blade in the femoral head-neck fragment led to the cut-out. Postoperative mobilization was performed according to the standard protocol of our department, starting with full weight-bearing at the second day post-surgery.

- One of the very first and to date few clinical papers on the use of modern generation nails in such fractures should be referred to and reported on references:

Carulli C, Piacentini F, Paoli T, Civinini R, Innocenti M. A comparison of two fixation methods for femoral trochanteric fractures: a new generation intramedullary system vs sliding hip screw. Clin Cases Miner Bone Metab. 2017 Jan-Apr;14(1):40-47. doi: 10.11138/ccmbm/2017.14.1.040. Epub 2017 May 30. PMID: 28740524

àchanges were made accordingly; literature was added to the manuscript.

Reviewer 2 Report

Thank you for the opportunity to review this interesting manuscript. It is a study to investigate the cut-out rates, migration of the lag screw, and implant failure after intramedullary treatment of AO 31.A2 pertrochanteric fractures. The study made a comparison of Gamma Nail with and without U-Blade and PFNA. The authors are to be congratulated on their findings and achievements on the comparisons of clinical and radiological results between the latest most used implant technologies in unstable AO/OTA 31. A2.1-3 femur pertrochanteric fractures.

In my opinion, the authors provided a sound scientific report. The objectives were clearly stated. The necessities of the study were adequately explained. The study method was adequately clarified. The results were clearly presented and the figures were clearly illustrated. The discussion pointed out the important findings. The conclusions appropriately based on the survey results and discussions.

I am appreciated for the fact and congratulated on authors’ achievement. however, some concerns had raised from this work. I will be appreciated if the authors would provide the title of each table illustrating the purpose of the table. The authors have stated the multiple regression analysis would be used to examine the independent associations of various demographic and injury-related factors. However, the results were not be presented. I will be appreciated if the authors would provide the results for this analysis and identify and discuss some key factors. Concerning clinical outcomes comparisons between groups, the authors only presented all patients’ outcome, no comparisons between groups were presented in the tables or the texts. Moreover, the results of the mean operation time were not presented. The major issues in the study are selection bias and preoperative parameters comparisons due to the retrospective study design as the authors stated in the limitation section. Moreover, whether “asses” at line 18, Page 1 might be corrected to “assess”.

Author Response

I am appreciated for the fact and congratulated on authors’ achievement. however, some concerns had raised from this work. I will be appreciated if the authors would provide the title of each table illustrating the purpose of the table. 

àchanges were made accordingly

The authors have stated the multiple regression analysis would be used to examine the independent associations of various demographic and injury-related factors. However, the results were not be presented. I will be appreciated if the authors would provide the results for this analysis and identify and discuss some key factors.

àParameters like age, gender, BMI, ASA Score and time of surgery had no influence on the likelihood of cut-out or sustaining a complication. Furthermore, we were just able to confirm a higher TAD distance as predictor of failure in the PFNA group, whereas in the Gamma and the U-Blade group the TAD and CalTAD did not predict the likelihood of cut-out.

Both the mean TAD and calTAD among those who suffered from cut-out was higher than those who were spared this complication, 20.0mm and 15.3mm vs 18.5mm and 13.1mm, respectively. Neither of these differences achieved statistical significance (p=n.s.).”

à“Several factors are associated with failure of cephalomedullary nailing for pertrochanteric fractures. Recent literature identified no relation between patient demographics and implant failure. However, increased TAD and unstable, or poorly reduced, fractures were found to be the principal causes of implant cut-out and failure. In keeping with this data, the single PFNA cut-out had a significantly higher TAD. Contrastingly, adequate fracture reduction was achieved in all our patients who subsequently suffered a cut-out.”

Concerning clinical outcomes comparisons between groups, the authors only presented all patients’ outcome, no comparisons between groups were presented in the tables or the texts.

àchanges were made accordingly, significant differences within the tables have been highlighted and are mentioned in the text.

Moreover, the results of the mean operation time were not presented. 

àchanges were made accordingly, we found a significant longer operation time for the U-Blade group (p<0.0040)< span="">

The major issues in the study are selection bias and preoperative parameters comparisons due to the retrospective study design as the authors stated in the limitation section.

Moreover, whether “asses” at line 18, Page 1 might be corrected to “assess”.

àchanges were made accordingly.

Proof reading of the manuscript was performed by an English native speaker.

Round  2

Reviewer 2 Report

Thank you for the opportunity to review this interesting manuscript. I am appreciated for the authors’ precious time to response to my concerns and revise the manuscript. I have no further comment. Thanks a lot.

J. Clin. Med. EISSN 2077-0383 Published by MDPI AG, Basel, Switzerland RSS E-Mail Table of Contents Alert
Back to Top